# A DIRT-T Approach to Unsupervised Domain Adaptation

**Rui Shu**[†][*]**, Hung H. Bui**[‡]**, Hirokazu Narui**[†]**, & Stefano Ermon**[†]

[†]Stanford University
[‡]DeepMind
[†]{ruishu,hirokaz2,ermon}@stanford.edu
[‡]{buih}@google.com

## Abstract

Domain adaptation refers to the problem of leveraging labeled data in a source domain to learn an accurate model in a target domain where labels are scarce or unavailable. A recent approach for finding a common representation of the two domains is via domain adversarial training (Ganin & Lempitsky, 2015), which attempts to induce a feature extractor that matches the source and target feature distributions in some feature space. However, domain adversarial training faces two critical limitations: 1) if the feature extraction function has high-capacity, then feature distribution matching is a weak constraint, 2) in non-conservative domain adaptation (where no single classifier can perform well in both the source and target domains), training the model to do well on the source domain hurts performance on the target domain. In this paper, we address these issues through the lens of the cluster assumption, i.e., decision boundaries should not cross high-density data regions. We propose two novel and related models: 1) the Virtual Adversarial Domain Adaptation (VADA) model, which combines domain adversarial training with a penalty term that punishes violation of the cluster assumption; 2) the Decision-boundary Iterative Refinement Training with a Teacher (DIRT-T)[1] model, which takes the VADA model as initialization and employs natural gradient steps to further minimize the cluster assumption violation. Extensive empirical results demonstrate that the combination of these two models significantly improve the state-of-the-art performance on the digit, traffic sign, and Wi-Fi recognition domain adaptation benchmarks.

## 1 Introduction

The development of deep neural networks has enabled impressive performance in a wide variety of machine learning tasks. However, these advancements often rely on the existence of a large amount of labeled training data. In many cases, direct access to vast quantities of labeled data for the task of interest (the target domain) is either costly or otherwise absent, but labels are readily available for related training sets (the source domain). A notable example of this scenario occurs when the source domain consists of richly-annotated synthetic or semi-synthetic data, but the target domain consists of unannotated real-world data (Sun & Saenko, 2014; Vazquez et al., 2014). However, the source data distribution is often dissimilar to the target data distribution, and the resulting significant covariate shift is detrimental to the performance of the source-trained model when applied to the target domain (Shimodaira, 2000).

Solving the covariate shift problem of this nature is an instance of domain adaptation (Ben-David et al., 2010b). In this paper, we consider a challenging setting of domain adaptation where 1) we are provided with fully-labeled source samples and completely-unlabeled target samples, and 2) the existence of a classifier in the hypothesis space with low generalization error in both source and target domains is not guaranteed. Borrowing approximately the terminology from Ben-David et al. (2010b), we refer to this setting as unsupervised, *non-conservative* domain adaptation. We note

---

[*]Work was done during first author's internship at Adobe Research.
[1]Pronounce as "dirty." Implementation available at https://github.com/RuiShu/dirt-t

that this is in contrast to *conservative* domain adaptation, where we assume our hypothesis space contains a classifier that performs well in both the source and target domains.

To tackle unsupervised domain adaptation, Ganin & Lempitsky (2015) proposed to constrain the classifier to only rely on domain-invariant features. This is achieved by training the classifier to perform well on the source domain while minimizing the divergence between features extracted from the source versus target domains. To achieve divergence minimization, Ganin & Lempitsky (2015) employ domain adversarial training. We highlight two issues with this approach: 1) when the feature function has high-capacity and the source-target supports are disjoint, the domain-invariance constraint is potentially very weak (see Section 3), and 2) good generalization on the source domain hurts target performance in the non-conservative setting.

Saito et al. (2017) addressed these issues by replacing domain adversarial training with asymmetric tri-training (ATT), which relies on the assumption that target samples that are labeled by a source-trained classifier with high confidence *are* correctly labeled by the source classifier. In this paper, we consider an orthogonal assumption: the cluster assumption (Chapelle & Zien, 2005), that the input distribution contains separated data clusters and that data samples in the same cluster share the same class label. This assumption introduces an additional bias where we seek decision boundaries that do not go through high-density regions. Based on this intuition, we propose two novel models: 1) the Virtual Adversarial Domain Adaptation (VADA) model which incorporates an additional virtual adversarial training (Miyato et al., 2017) and conditional entropy loss to push the decision boundaries away from the empirical data, and 2) the Decision-boundary Iterative Refinement Training with a Teacher (DIRT-T) model which uses natural gradients to further refine the output of the VADA model while focusing purely on the target domain. We demonstrate that

1. In conservative domain adaptation, where the classifier is trained to perform well on the source domain, VADA can be used to further constrain the hypothesis space by penalizing violations of the cluster assumption, thereby improving domain adversarial training.

2. In non-conservative domain adaptation, where we account for the mismatch between the source and target optimal classifiers, DIRT-T allows us to transition from a joint (source and target) classifier (VADA) to a better target domain classifier. Interestingly, we demonstrate the advantage of natural gradients in DIRT-T refinement steps.

We report results for domain adaptation in digits classification (MNIST-M, MNIST, SYN DIGITS, SVHN), traffic sign classification (SYN SIGNS, GTSRB), general object classification (STL-10, CIFAR-10), and Wi-Fi activity recognition (Yousefi et al., 2017). We show that, in nearly all experiments, VADA improves upon previous methods and that DIRT-T improves upon VADA, setting new state-of-the-art performances across a wide range of domain adaptation benchmarks. In adapting MNIST $\rightarrow$ SVHN, a very challenging task, we out-perform ATT by over $20\%$.

## 2  RELATED WORK

Given the extensive literature on domain adaptation, we highlight several works most relevant to our paper. Shimodaira (2000); Mansour et al. (2009) proposed to correct for covariate shift by re-weighting the source samples such that the discrepancy between the target distribution and re-weighted source distribution is minimized. Such a procedure is problematic, however, if the source and target distributions do not contain sufficient overlap. Huang et al. (2007); Long et al. (2015); Ganin & Lempitsky (2015) proposed to instead project both distributions into some feature space and encourage distribution matching in the feature space. Ganin & Lempitsky (2015) in particular encouraged feature matching via domain adversarial training, which corresponds approximately to Jensen-Shannon divergence minimization (Goodfellow et al., 2014). To better perform non-conservative domain adaptation, Saito et al. (2017) proposed to modify tri-training (Zhou & Li, 2005) for domain adaptation, leveraging the assumption that highly-confident predictions are correct predictions (Zhu, 2005). Several of aforementioned methods are based on Ben-David et al. (2010a)'s theoretical analysis of domain adaptation, which states the following,

**Theorem 1** *(Ben-David et al., 2010a) Let $\mathcal{H}$ be the hypothesis space and let $(X_s, \epsilon_s)$ and $(X_t, \epsilon_t)$ be the two domains and their corresponding generalization error functions. Then for any $h \in \mathcal{H}$,*

$$\epsilon_t(h) \leq \frac{1}{2} d_{\mathcal{H}\Delta\mathcal{H}}(X_s, X_t) + \epsilon_s(h) + \min_{h' \in \mathcal{H}} \epsilon_t(h') + \epsilon_s(h'), \tag{1}$$

*where $d_{\mathcal{H}\Delta\mathcal{H}}$ denotes the $\mathcal{H}\Delta\mathcal{H}$-distance between the domains $X_s$ and $X_t$,*

$$d_{\mathcal{H}\Delta\mathcal{H}} = 2 \sup_{h, h' \in \mathcal{H}} \left| \mathbb{E}_{x \sim X_s}[h(x) \neq h'(x)] - \mathbb{E}_{x \sim X_t}[h(x) \neq h'(x)] \right|. \tag{2}$$

Intuitively, $d_{\mathcal{H}\Delta\mathcal{H}}$ measures the extent to which small changes to the hypothesis in the source domain can lead to large changes in the target domain. It is evident that $d_{\mathcal{H}\Delta\mathcal{H}}$ relates intimately to the complexity of the hypothesis space and the divergence between the source and target domains. For infinite-capacity models and domains with disjoint supports, $d_{\mathcal{H}\Delta\mathcal{H}}$ is maximal.

A critical component to our paper is the cluster assumption, which states that decision boundaries should not cross high-density regions (Chapelle & Zien, 2005). This assumption has been extensively studied and leveraged for semi-supervised learning, leading to proposals such as conditional entropy minimization (Grandvalet & Bengio, 2005) and pseudo-labeling (Lee, 2013). More recently, the cluster assumption has led to many successful deep semi-supervised learning algorithms such as semi-supervised generative adversarial networks (Dai et al., 2017), virtual adversarial training (Miyato et al., 2017), and self/temporal-ensembling (Laine & Aila, 2016; Tarvainen & Valpola, 2017). Given the success of the cluster assumption in semi-supervised learning, it is natural to consider its application to domain adaptation. Indeed, Ben-David & Urner (2014) formalized the cluster assumption through the lens of probabilistic Lipschitzness and proposed a nearest-neighbors model for domain adaptation. Our work extends this line of research by showing that the cluster assumption can be applied to deep neural networks to solve complex, high-dimensional domain adaptation problems. Independently of our work, French et al. (2017) demonstrated the application of self-ensembling to domain adaptation. However, our work additionally considers the application of the cluster assumption to non-conservative domain adaptation.

## 3 LIMITATION OF DOMAIN ADVERSARIAL TRAINING

Before describing our model, we first highlight that domain adversarial training may not be sufficient for domain adaptation if the feature extraction function has high-capacity. Consider a classifier $h_\theta$, parameterized by $\theta$, that maps inputs to the $(K-1)$-simplex (denote as $\mathcal{C}$), where $K$ is the number of classes. Suppose the classifier $h = g \circ f$ can be decomposed as the composite of an embedding function $f_\theta : \mathcal{X} \rightarrow \mathcal{Z}$ and embedding classifier $g_\theta : \mathcal{Z} \rightarrow \mathcal{C}$. For the source domain, let $\mathcal{D}_s$ be the joint distribution over input $x$ and one-hot label $y$ and let $X_s$ be the marginal input distribution. $(\mathcal{D}_t, X_t)$ are analogously defined for the target domain. Let $(\mathcal{L}_s, \mathcal{L}_d)$ be the loss functions

$$\mathcal{L}_y(\theta; \mathcal{D}_s) = \mathbb{E}_{x, y \sim \mathcal{D}_s} \left[ y^\top \ln h_\theta(x) \right] \tag{3}$$

$$\mathcal{L}_d(\theta; \mathcal{D}_s, \mathcal{D}_t) = \sup_D \mathbb{E}_{x \sim \mathcal{D}_s} \left[ \ln D(f_\theta(x)) \right] + \mathbb{E}_{x \sim \mathcal{D}_t} \left[ \ln(1 - D(f_\theta(x))) \right], \tag{4}$$

where the supremum ranges over discriminators $D : \mathcal{Z} \rightarrow (0, 1)$. Then $\mathcal{L}_y$ is the cross-entropy objective and $D$ is a domain discriminator. Domain adversarial training minimizes the objective

$$\min_\theta \mathcal{L}_y(\theta; \mathcal{D}_s) + \lambda_d \mathcal{L}_d(\theta; \mathcal{D}_s, \mathcal{D}_t), \tag{5}$$

where $\lambda_d$ is a weighting factor. Minimization of $\mathcal{L}_d$ encourages the learning of a feature extractor $f$ for which the Jensen-Shannon divergence between $f(X_s)$ and $f(X_t)$ is small.[2] Ganin & Lempitsky (2015) suggest that successful adaptation tends to occur when the source generalization error and feature divergence are both small.

It is easy, however, to construct situations where this suggestion fails. In particular, if $f$ has infinite-capacity and the source-target supports are disjoint, then $f$ can employ arbitrary transformations to the target domain so as to match the source feature distribution (see Appendix E for formalization).

---

[2]In practice, the minimization of $\mathcal{L}_d$ requires solving a mini-max optimization problem. We discuss this in more detail in Appendix C

We verify empirically that, for sufficiently deep layers, jointly achieving small source generalization error and feature divergence does not imply high accuracy on the target task (Table 5). Given the limitations of domain adversarial training, we wish to identify additional constraints that one can place on the model to achieve better, more reliable domain adaptation.

# 4 CONSTRAINING VIA CONDITIONAL ENTROPY MINIMIZATION

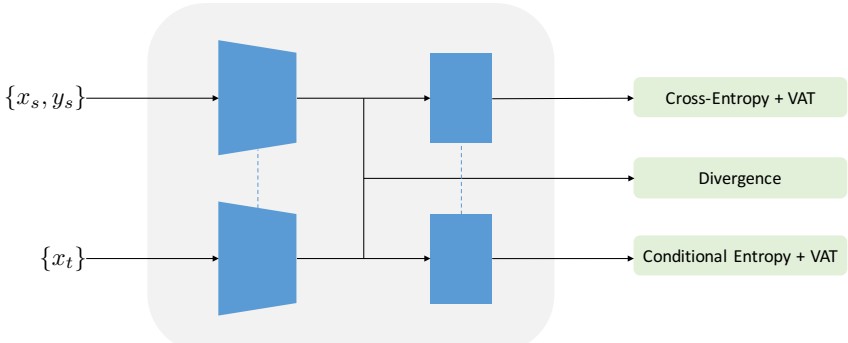

Figure 1: VADA improves upon domain adversarial training by additionally penalizing violations of the cluster assumption.

In this paper, we apply the cluster assumption to domain adaptation. The cluster assumption states that the input distribution $X$ contains clusters and that points in the same cluster come from the same class. This assumption has been extensively studied and applied successfully to a wide range of classification tasks (see Section 2). If the cluster assumption holds, the optimal decision boundaries should occur far away from data-dense regions in the space of $\mathcal{X}$ (Chapelle & Zien, 2005). Following Grandvalet & Bengio (2005), we achieve this behavior via minimization of the conditional entropy with respect to the target distribution,

$$\mathcal{L}_c(\theta; \mathcal{D}_t) = -\mathbb{E}_{x \sim \mathcal{D}_t} \left[ h_\theta(x)^\top \ln h_\theta(x) \right]. \tag{6}$$

Intuitively, minimizing the conditional entropy forces the classifier to be confident on the unlabeled target data, thus driving the classifier's decision boundaries away from the target data (Grandvalet & Bengio, 2005). In practice, the conditional entropy must be empirically estimated using the available data. However, Grandvalet & Bengio (2005) note that this approximation breaks down if the classifier $h$ is not locally-Lipschitz. Without the locally-Lipschitz constraint, the classifier is allowed to abruptly change its prediction in the vicinity of the training data points, which 1) results in a unreliable empirical estimate of conditional entropy and 2) allows placement of the classifier decision boundaries close to the training samples even when the empirical conditional entropy is minimized. To prevent this, we propose to explicitly incorporate the locally-Lipschitz constraint via virtual adversarial training (Miyato et al., 2017) and add to the objective function the additional term

$$\mathcal{L}_v(\theta; \mathcal{D}) = \mathbb{E}_{x \sim \mathcal{D}} \left[ \max_{\|r\| \leq \epsilon} \mathrm{D}_{\mathrm{KL}}(h_\theta(x) \| h_\theta(x + r)) \right], \tag{7}$$

which enforces classifier consistency within the norm-ball neighborhood of each sample $x$. Note that virtual adversarial training can be applied with respect to either the target or source distributions. We can combine the conditional entropy minimization objective and domain adversarial training to yield

$$\min_\theta \mathcal{L}_y(\theta; \mathcal{D}_s) + \lambda_d \mathcal{L}_d(\theta; \mathcal{D}_s, \mathcal{D}_t) + \lambda_s \mathcal{L}_v(\theta; \mathcal{D}_s) + \lambda_t \left[ \mathcal{L}_v(\theta; \mathcal{D}_t) + \mathcal{L}_c(\theta; \mathcal{D}_t) \right], \tag{8}$$

a basic combination of domain adversarial training and semi-supervised training objectives. We refer to this as the Virtual Adversarial Domain Adaptation (VADA) model. Empirically, we observed that the hyperparameters $(\lambda_d, \lambda_s, \lambda_t)$ are easy to choose and work well across multiple tasks (Appendix B).

$\mathcal{H}\Delta\mathcal{H}$-**Distance Minimization**. VADA aligns well with the theory of domain adaptation provided in Theorem 1. Let the loss,

$$\mathcal{L}_t(\theta) = \mathcal{L}_v(\theta; \mathcal{D}_t) + \mathcal{L}_c(\theta; D_t), \tag{9}$$

denote the degree to which the target-side cluster assumption is violated. Modulating $\lambda_t$ enables VADA to trade-off between hypotheses with low target-side cluster assumption violation and hypotheses with low source-side generalization error. Setting $\lambda_t > 0$ allows rejection of hypotheses with high target-side cluster assumption violation. By rejecting such hypotheses from the hypothesis space $\mathcal{H}$, VADA reduces $d_{\mathcal{H}\Delta\mathcal{H}}$ and yields a tighter bound on the target generalization error. We verify empirically that VADA achieves significant improvements over existing models on multiple domain adaptation benchmarks (Table 1).

## 5 DECISION-BOUNDARY ITERATIVE REFINEMENT TRAINING

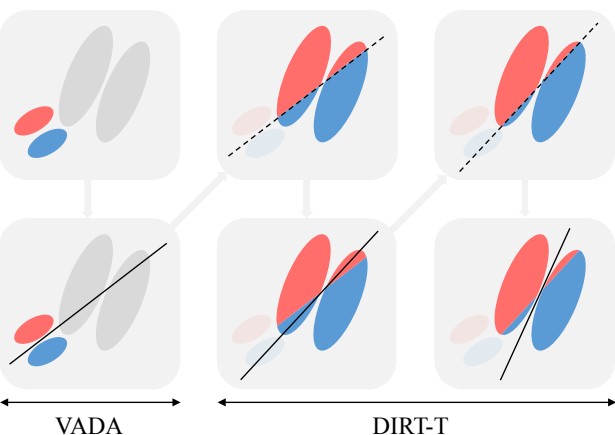

Figure 2: DIRT-T uses VADA as initialization. After removing the source training signal, DIRT-T minimizes cluster assumption violation in the target domain through a series of natural gradient steps.

In non-conservative domain adaptation, we assume the following inequality,

$$\min_{h \in \mathcal{H}} \epsilon_t(h) < \epsilon_t(h^a) \text{ where } h^a = \arg\min_{h \in \mathcal{H}} \epsilon_s(h) + \epsilon_t(h), \tag{10}$$

where $(\epsilon_s, \epsilon_t)$ are generalization error functions for the source and target domains. This means that, for a given hypothesis class $\mathcal{H}$, the optimal classifier in the source domain does not coincide with the optimal classifier in the target domain.

We assume that the optimality gap in Eq. (10) results from violation of the cluster assumption. In other words, we suppose that any source-optimal classifier drawn from our hypothesis space *necessarily* violates the cluster assumption in the target domain. Insofar as VADA is trained on the source domain, we hypothesize that a better hypothesis is achievable by introducing a secondary training phase that solely minimizes the target-side cluster assumption violation.

Under this assumption, the natural solution is to initialize with the VADA model and then further minimize the cluster assumption violation in the target domain. In particular, we first use VADA to learn an initial classifier $h_{\theta_0}$. Next, we incrementally push the classifier's decision boundaries away from data-dense regions by minimizing the target-side cluster assumption violation loss $\mathcal{L}_t$ in Eq. (9). We denote this procedure Decision-boundary Iterative Refinement Training (DIRT).

### 5.1 DECISION-BOUNDARY ITERATIVE REFINEMENT TRAINING WITH A TEACHER

Stochastic gradient descent minimizes the loss $\mathcal{L}_t$ by selecting gradient steps $\Delta\theta$ according to the following objective,

$$\min_{\Delta\theta}. \mathcal{L}_t(\theta + \Delta\theta) \tag{11}$$

$$\text{s.t. } \|\Delta\theta\| \leq \epsilon, \tag{12}$$

which defines the neighborhood in the parameter space. This notion of neighborhood is sensitive to the parameterization of the model; depending on the parameterization, a seemingly small step $\Delta\theta$ may result in a vastly different classifier. This contradicts our intention of incrementally and locally pushing the decision boundaries to a local conditional entropy minimum, which requires that the decision boundaries of $h_{\theta+\Delta\theta}$ stay close to that of $h_\theta$. It is therefore important to define a neighborhood that is parameterization-invariant. Following Pascanu & Bengio (2013), we instead select $\Delta\theta$ using the following objective,

$$\min_{\Delta\theta} \mathcal{L}_t(\theta + \Delta\theta)$$
$$\text{s.t. } \mathbb{E}_{x \sim D_t} \left[ D_{\mathrm{KL}}(h_\theta(x) \| h_{\theta+\Delta\theta}(x)) \right] \leq \epsilon. \tag{13}$$

Each optimization step now solves for a gradient step $\Delta\theta$ that minimizes the conditional entropy, subject to the constraint that the Kullback-Leibler divergence between $h_\theta(x)$ and $h_{\theta+\Delta\theta}(x)$ is small for $x \sim \mathcal{X}_t$. The corresponding Lagrangian suggests that one can instead minimize a sequence of optimization problems

$$\min_{\theta_n} \lambda_t \mathcal{L}_t(\theta_n) + \beta_t \mathbb{E} \left[ D_{\mathrm{KL}}(h_{\theta_{n-1}}(x) \| h_{\theta_n}(x)) \right], \tag{14}$$

that approximates the application of a series of natural gradient steps.

In practice, each of the optimization problems in Eq. (14) can be solved approximately via a finite number of stochastic gradient descent steps. We denote the number of steps taken to be the refinement interval $B$. Similar to Tarvainen & Valpola (2017), we use the Adam Optimizer with Polyak averaging (Polyak & Juditsky, 1992). We interpret $h_{\theta_{n-1}}$ as a (sub-optimal) teacher for the student model $h_{\theta_n}$, which is trained to stay close to the teacher model while seeking to reduce the cluster assumption violation. As a result, we denote this model as Decision-boundary Iterative Refinement Training with a Teacher (DIRT-T).

**Weakly-Supervised Learning**. This sequence of optimization problems has a natural interpretation that exposes a connection to weakly-supervised learning. In each optimization problem, the teacher model $h_{\theta_{n-1}}$ pseudo-labels the target samples with noisy labels. Rather than naively training the student model $h_{\theta_n}$ on the noisy labels, the additional training signal $\mathcal{L}_t$ allows the student model to place its decision boundaries further from the data. If the clustering assumption holds and the initial noisy labels are sufficiently similar to the true labels, conditional entropy minimization can improve the placement of the decision boundaries (Reed et al., 2014).

**Domain Adaptation**. An alternative interpretation is that DIRT-T is the *recursive* extension of VADA, where the act of pseudo-labeling of the target distribution constructs a new "source" domain (i.e. target distribution $X_t$ with pseudo-labels). The sequence of optimization problems can then be seen as a sequence of non-conservative domain adaptation problems in which $X_s = X_t$ but $p_s(y \mid x) \neq p_t(y \mid x)$, where $p_s(y \mid x) = h_{\theta_{n-1}}(x)$ and $p_t(y \mid x)$ is the true conditional label distribution in the target domain. Since $d_{\mathcal{H}\Delta\mathcal{H}}$ is strictly zero in this sequence of optimization problems, domain adversarial training is no longer necessary. Furthermore, if $\mathcal{L}_t$ minimization does improve the student classifier, then the gap in Eq. (10) should get smaller each time the source domain is updated.

## 6 Experiments

In principle, our method can be applied to any domain adaptation tasks so long as one can define a reasonable notion of neighborhood for virtual adversarial training (Miyato et al., 2016). For comparison against Saito et al. (2017) and French et al. (2017), we focus on visual domain adaptation and evaluate on MNIST, MNIST-M, Street View House Numbers (SVHN), Synthetic Digits (SYN DIGITS), Synthetic Traffic Signs (SYN SIGNS), the German Traffic Signs Recognition Benchmark (GTSRB), CIFAR-10, and STL-10. For non-visual domain adaptation, we evaluate on Wi-Fi activity recognition.

### 6.1 Implementation Detail

**Architecture** We use a small CNN for the digits, traffic sign, and Wi-Fi domain adaptation experiments, and a larger CNN for domain adaptation between CIFAR-10 and STL-10. Both architectures are available in Appendix A. For fair comparison, we additionally report the performance of

source-only baseline models and demonstrate that the significant improvements are attributable to our proposed method.

**Replacing gradient reversal**. In contrast to Ganin & Lempitsky (2015), which proposed to implement domain adversarial training via gradient reversal, we follow Goodfellow et al. (2014) and instead optimize via alternating updates to the discriminator and encoder (see Appendix C).

**Instance normalization**. We explored the application of instance normalization as an image preprocessing step. This procedure makes the classifier invariant to channel-wide shifts and rescaling of pixel intensities. A discussion of instance normalization for domain adaptation is provided in Appendix D. We show in Figure 3 the effect of applying instance normalization to the input image.

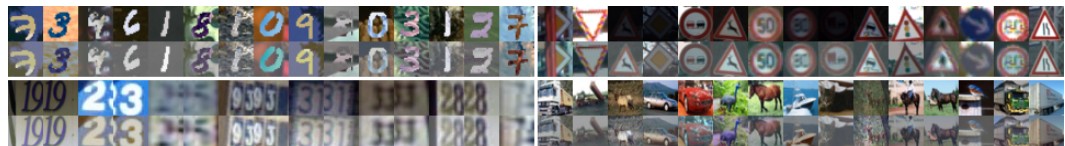

Figure 3: Effect of applying instance normalization to the input image. In clockwise direction: MNIST-M, GTSRB, SVHN, and CIFAR-10. In each quadrant, the top row is the original image, and the bottom row is the instance-normalized image.

**Hyperparameters**. For each task, we tuned the four hyperparameters $(\lambda_d, \lambda_s, \lambda_t, \beta)$ by randomly selecting 1000 labeled target samples from the training set and using that as our validation set. We observed that extensive hyperparameter-tuning is not necessary to achieve state-of-the-art performance. In all experiments with instance-normalized inputs, we restrict our hyperparameter search for each task to $\lambda_d = \{0, 10^{-2}\}, \lambda_s = \{0, 1\}, \lambda_t = \{10^{-2}, 10^{-1}\}$. We fixed $\beta = 10^{-2}$. Note that the decision to turn $(\lambda_d, \lambda_s)$ on or off that can often be determined *a priori*. A complete list of the hyperparameters is provided in Appendix B.

## 6.2 MODEL EVALUATION

| Source
Target | MNIST
MNIST-M | SVHN
MNIST | MNIST
SVHN | DIGITS
SVHN | SIGNS
GTSRB | CIFAR
STL | STL
CIFAR |
|---|---|---|---|---|---|---|---|
| MMD (Long et al., 2015) | 76.9 | 71.1 | - | 88.0 | 91.1 | - | - |
| DANN (Ganin & Lempitsky, 2015) | 81.5 | 71.1 | 35.7 | 90.3 | 88.7 | - | - |
| DRCN (Ghifary et al., 2016) | - | 82.0 | 40.1 | - | - | 66.4 | 58.7 |
| DSN (Bousmalis et al., 2016b) | 83.2 | 82.7 | - | 91.2 | 93.1 | - | - |
| kNN-Ad (Sener et al., 2016) | 86.7 | 78.8 | 40.3 | - | - | - | - |
| PixelDA (Bousmalis et al., 2016a) | 98.2 | - | - | - | - | - | - |
| ATT (Saito et al., 2017) | 94.2 | 86.2 | 52.8 | 92.9 | 96.2 | - | - |
| Π-model (aug) (French et al., 2017) | - | 92.0 | 71.4 | 94.2 | 98.4 | 76.3 | 64.2 |
| *Without Instance-Normalized Input:* | | | | | | | |
| Source-Only | 58.5 | 77.0 | 27.9 | 86.9 | 79.6 | 76.3 | 63.6 |
| VADA | 97.7 | 97.9 | 47.5 | 94.8 | 98.8 | **80.0** | 73.5 |
| DIRT-T | **98.9** | **99.4** | **54.5** | **96.1** | **99.5** | - | **75.3** |
| *With Instance-Normalized Input:* | | | | | | | |
| Source-Only | 59.9 | 82.4 | 40.9 | 88.6 | 86.2 | 77.0 | 62.6 |
| VADA | 95.7 | 94.5 | 73.3 | 94.9 | 99.2 | **78.3** | 71.4 |
| DIRT-T | **98.7** | **99.4** | **76.5** | **96.2** | **99.6** | - | **73.3** |

Table 1: Test set accuracy on visual domain adaptation benchmarks. In all settings, both VADA and DIRT-T achieve state-of-the-art performance in all settings.

**MNIST → MNIST-M**. We first evaluate the adaptation from MNIST to MNIST-M. MNIST-M is constructed by blending MNIST digits with random color patches from the BSDS500 dataset.

**MNIST ↔ SVHN**. The distribution shift is exacerbated when adapting between MNIST and SVHN. Whereas MNIST consists of black-and-white handwritten digits, SVHN consists of crops of colored, street house numbers. Because MNIST has a significantly lower intrinsic dimensionality that SVHN, the adaptation from MNIST → SVHN is especially challenging when the input is not pre-processed

via instance normalization. When instance normalization is applied, we achieve a strong state-of-the-art performance $76.5\%$ and an equally impressive margin-of-improvement over source-only of $35.6\%$. Interestingly, by reducing the refinement interval $B$ and taking noisier natural gradient steps, we were occasionally able to achieve accuracies as high as $87\%$. However, due to the high-variance associated with this, we omit reporting this configuration in Table 1.

**SYN DIGITS → SVHN**. The adaptation from SYN DIGITS → SVHN reflect a common adaptation problem of transferring from synthetic images to real images. The SYN DIGITS dataset consist of $500000$ images generated from Windows fonts by varying the text, positioning, orientation, background, stroke color, and the amount of blur.

**SYN SIGNS → GTSRB**. This setting provides an additional demonstration of adapting from synthetic images to real images. Unlike SYN DIGITS → SVHN, SYN SIGNS → GTSRB contains 43 classes instead of 10.

**STL ↔ CIFAR**. Both STL-10 and CIFAR-10 are 10-class image datasets. These two datasets contain nine overlapping classes. Following the procedure in French et al. (2017), we removed the non-overlapping classes ("frog" and "monkey") and reduce to a 9-class classification problem. We achieve state-of-the-art performance in both adaptation directions. In STL → CIFAR, we achieve a $11.7\%$ margin-of-improvement and a performance accuracy of $73.3\%$. Note that because STL-10 contains a very small training set, it is difficult to estimate the conditional entropy, thus making DIRT-T unreliable for CIFAR → STL.

| Source | Room A |
| --- | --- |
| Target | Room B |
| With Instance-Normalized Input: | |
| Source-Only | 35.7 |
| DANN | 38.0 |
| VADA | **53.0** |
| DIRT-T | **53.0** |

Table 2: Results of the domain adaptation experiments on Wi-Fi Activity Recognition Task

**Wi-Fi Activity Recognition**. To evaluate the performance of our models on a non-visual domain adaptation task, we applied VADA and DIRT-T to the Wi-Fi Activity Recognition Dataset (Yousefi et al., 2017). The Wi-Fi Activity Recognition Dataset is a classification task that takes the Wi-Fi Channel State Information (CSI) data stream as input $x$ to predict motion activity within an indoor area as output $y$. Domain adaptation is necessary when the training and testing data are collected from different rooms, which we denote as Rooms A and B. Table 2 shows that VADA significantly improves classification accuracy compared to Source-Only and DANN by $17.3\%$ and $15\%$ respectively. However, DIRT-T does not lead to further improvements on this dataset. We perform experiments in Appendix F which suggests that VADA already achieves strong clustering in the target domain for this dataset, and therefore DIRT-T is not expected to yield further performance improvement.

| Source | MNIST | SVHN | MNIST | DIGITS | SIGNS | CIFAR | STL |
| --- | --- | --- | --- | --- | --- | --- | --- |
| Target | MNIST-M | MNIST | SVHN | SVHN | GTSRB | STL | CIFAR |
| ATT | 37.1 | 16.1 | 17.9 | 9.0 | **20.5** | - | - |
| Π-model (aug) | - | 3.7 | 18.1 | **10.6** | 1.0 | **4.5** | 7.4 |
| DIRT-T | **40.4** | **22.4** | 26.6 | 9.2 | 19.9 | - | **11.7** |
| DIRT-T (W.I.N.I.) | 38.8 | 17.0 | **35.6** | 7.6 | 13.4 | - | 10.7 |

Table 3: Additional comparison of the margin of improvement computed by taking the reported performance of each model and subtracting the reported source-only performance in the respective papers. W.I.N.I. indicates "with instance-normalized input."

**Overall**. We achieve state-of-the-art results across all tasks. For a fairer comparison against ATT and the Π-model, Table 3 provides the improvement margin over the respective source-only performance reported in each paper. In four of the tasks (MNIST → MNIST-M, SVHN → MNIST, MNIST → SVHN, STL → CIFAR), we achieve substantial margin of improvement compared to previous models. In the remaining three tasks, our improvement margin over the source-only model

is competitive against previous models. Our closest competitor is the Π-model. However, unlike the Π-model, we do not perform data augmentation.

It is worth noting that DIRT-T consistently improves upon VADA. Since DIRT-T operates by incrementally pushing the decision boundaries away from the target domain data, it relies heavily on the cluster assumption. DIRT-T's empirical success therefore demonstrates the effectiveness of leveraging the cluster assumption in unsupervised domain adaptation with deep neural networks.

## 6.3 ANALYSIS OF VADA AND DIRT-T

### 6.3.1 ROLE OF VIRTUAL ADVERSARIAL TRAINING

To study the relative contribution of the virtual adversarial training in the VADA and DIRT-T objectives (Eq. (8) and Eq. (14) respectively), we perform an extensive ablation analysis in Table 4. The removal of the virtual adversarial training component is denoted by the "no-vat" subscript. Our results show that $VADA_{no\text{-}vat}$ is sufficient for out-performing DANN in all but one task. The further ability for $DIRT\text{-}T_{no\text{-}vat}$ to improve upon $VADA_{no\text{-}vat}$ demonstrates the effectiveness of conditional entropy minimization. Ultimately, in six of the seven tasks, both virtual adversarial training and conditional entropy minimization are essential for achieving the best performance. The empirical importance of incorporating virtual adversarial training shows that the locally-Lipschitz constraint is beneficial for pushing the classifier decision boundaries away from data.

| Source
Target | MNIST
MNIST-M | SVHN
MNIST | MNIST
SVHN | DIGITS
SVHN | SIGNS
GTSRB | CIFAR
STL | STL
CIFAR |
|---|---|---|---|---|---|---|---|
| With Instance-Normalized Input: | | | | | | | |
| Source-Only | 59.9 | 82.4 | 40.9 | 88.6 | 86.2 | 77.0 | 62.6 |
| DANN (our implementation) | 94.6 | 68.3 | 60.6 | 90.1 | 97.5 | 78.1 | 62.7 |
| $VADA_{no\text{-}vat}$ | 93.8 | 83.1 | 66.8 | 93.4 | 98.4 | **79.1** | 68.6 |
| $VADA_{no\text{-}vat} \rightarrow DIRT\text{-}T_{no\text{-}vat}$ | 94.8 | 96.3 | 68.6 | 94.4 | 99.1 | - | 69.2 |
| $VADA_{no\text{-}vat} \rightarrow DIRT\text{-}T$ | **98.3** | **99.4** | **69.8** | **95.3** | **99.6** | - | **71.0** |
| VADA | 95.7 | 94.5 | 73.3 | 94.9 | 99.2 | **78.3** | 71.4 |
| VADA $\rightarrow$ DIRT-T | **98.7** | **99.4** | **76.5** | **96.2** | **99.6** | - | **73.3** |

Table 4: Test set accuracy in ablation experiments, starting from the DANN model. The "no-vat" subscript denote models where the virtual adversarial training component is removed.

### 6.3.2 ROLE OF TEACHER MODEL IN DIRT-T

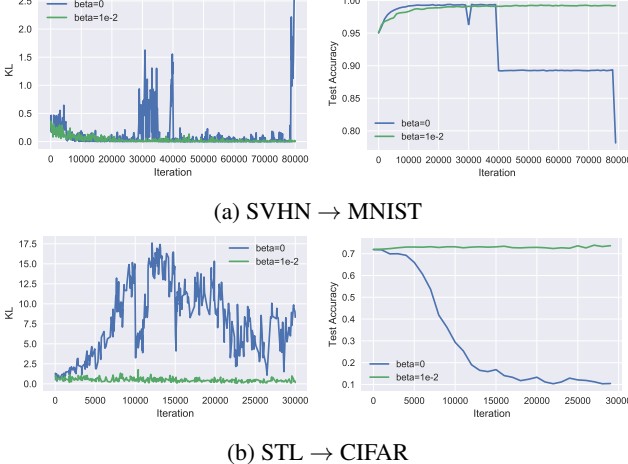

(a) SVHN → MNIST

(b) STL → CIFAR

Figure 4: Comparing model behavior with and without the application of the KL-term. At iteration 0, we begin with the VADA initialization and apply the DIRT-T algorithm.

When considering Eq. (14), it is natural to ask whether defining the neighborhood with respect to the classifier is truly necessary. In Figure 4, we demonstrate in SVHN → MNIST and STL → CIFAR that removal of the KL-term negatively impacts the model. Since the MNIST data manifold is low-dimensional and contains easily identifiable clusters, applying naive gradient descent (Eq. (12)) can also boost the test accuracy during initial training. However, without the KL constraint, the classifier can sometimes deviate significantly from the neighborhood of the previous classifier, and the resulting spikes in the KL-term correspond to sharp drops in target test accuracy. In STL → CIFAR, where the data manifold is much more complex and contains less obvious clusters, naive gradient descent causes immediate decline in the target test accuracy.

### 6.3.3 VISUALIZATION OF REPRESENTATION

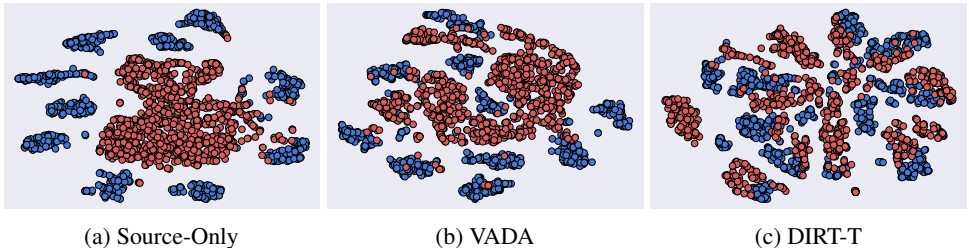

|   (a) Source-Only   |   (b) VADA   |   (c) DIRT-T   |

Figure 5: T-SNE plot of the last hidden layer for MNIST (blue) → SVHN (red). We used the model without instance normalization to highlight the further improvement that DIRT-T provides.

We further analyze the behavior of VADA and DIRT-T by showing T-SNE embeddings of the last hidden layer of the model trained to adapt from MNIST → SVHN. In Figure 5, source-only training shows strong clustering of the MNIST samples (blue) and performs poorly on SVHN (red). VADA offers significant improvement and exhibits signs of clustering on SVHN. DIRT-T begins with the VADA initialization and further enhances the clustering, resulting in the best performance on MNIST → SVHN.

### 6.4 DOMAIN ADVERSARIAL TRAINING: LAYER ABLATION

| | | DANN | | | VADA | |
|---|---|---|---|---|---|---|
| Layer | JSD $\geq$ | Source Accuracy | Target Accuracy | JSD $\geq$ | Source Accuracy | Target Accuracy |
| $L - 0$ | 0.001 | 78.0 | 24.7 | 0.001 | 24.9 | 18.4 |
| $L - 1$ | 0.002 | 98.6 | 35.0 | 0.007 | 12.0 | 11.6 |
| $L - 2$ | 0.353 | 16.4 | 10.3 | 0.383 | 11.5 | 9.9 |
| $L - 3$ | 0.036 | 94.8 | 33.8 | 0.034 | 67.8 | 37.1 |
| $L - 4$ | 0.012 | 97.0 | 40.0 | 0.020 | 96.8 | 61.5 |
| $L - 5$ | 0.235 | 99.3 | 57.9 | 0.244 | 99.4 | 73.3 |
| $L - 6$ | 0.486 | 99.2 | 60.3 | 0.509 | 99.3 | 70.4 |
| $L - 7$ | 0.644 | 99.0 | 52.5 | 0.608 | 99.1 | 70.5 |

Table 5: Comparison of model behavior when domain adversarial training is applied to various layers. We denote the very last (simplex) layer of the neural network as $L$ and ablatively domain adversarial training to the last eight layers. A lower bound on the Jensen-Shannon Divergence is computed by training a logistic regression model to predict domain origin when given the layer embeddings.

In Table 5, we applied domain adversarial training to various layers of a Domain Adversarial Neural Network (Ganin & Lempitsky, 2015) trained to adapt MNIST → SVHN. With the exception of layers $L - 2$ and $L - 0$, which experienced training instability, the general observation is that as the layer gets deeper, the additional capacity of the corresponding embedding function allows better matching of the source and target distributions without hurting source generalization accuracy. This demonstrates that the combination of low divergence and high source accuracy does not imply better adaptation to the target domain. Interestingly, when the classifier is regularized to be locally-Lipschitz via VADA, the combination of low divergence and high source accuracy appears to correlate more strongly with better adaptation.

## 7 CONCLUSION

In this paper, we presented two novel models for domain adaptation inspired by the cluster assumption. Our first model, VADA, performs domain adversarial training with an added term that penalizes violations of the cluster assumption. Our second model, DIRT-T, is an extension of VADA that recursively refines the VADA classifier by untethering the model from the source training signal and applying approximate natural gradients to further minimize the cluster assumption violation. Our experiments demonstrate the effectiveness of the cluster assumption: VADA achieves strong performance across several domain adaptation benchmarks, and DIRT-T further improves VADA performance. Our proposed models open up several possibilities for future work. One possibility is to apply DIRT-T to weakly supervised learning; another is to improve the natural gradient approximation via K-FAC (Martens & Grosse, 2015) and PPO (Schulman et al., 2017). Given the strong performance of our models, we also recommend them for other downstream domain adaptation applications.

ACKNOWLEDGMENTS

We gratefully acknowledge funding from Adobe, NSF (grants #1651565, #1522054, #1733686), Toyota Research Institute, Future of Life Institute, and Intel. We also thank Daniel Levy, Shengjia Zhao, and Jiaming Song for insightful discussions, and the anonymous reviewers for their helpful comments and suggestions.

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

## A ARCHITECTURES

| Layer Index | Small CNN | Large CNN |
|---|---|---|
| $L - 18$ | $32 \times 32 \times 3$ Image | |
| $L - 17$ | Instance Normalization (optional) | |
| $L - 16$ | $3 \times 3$ conv. 64 lReLU | $3 \times 3$ conv. 96 lReLU |
| $L - 15$ | $3 \times 3$ conv. 64 lReLU | $3 \times 3$ conv. 96 lReLU |
| $L - 14$ | $3 \times 3$ conv. 64 lReLU | $3 \times 3$ conv. 96 lReLU |
| $L - 13$ | 2 x 2 max-pool, stride 2 | |
| $L - 12$ | dropout, $p = 0.5$ | |
| $L - 11$ | Gaussian noise, $\sigma = 1$ | |
| $L - 10$ | $3 \times 3$ conv. 64 lReLU | $3 \times 3$ conv. 192 lReLU |
| $L - 9$ | $3 \times 3$ conv. 64 lReLU | $3 \times 3$ conv. 192 lReLU |
| $L - 8$ | $3 \times 3$ conv. 64 lReLU | $3 \times 3$ conv. 192 lReLU |
| $L - 7$ | 2 x 2 max-pool, stride 2 | |
| $L - 6$ | dropout, $p = 0.5$ | |
| $L - 5$ | Gaussian noise, $\sigma = 1$ | |
| $L - 4$ | $3 \times 3$ conv. 64 lReLU | $3 \times 3$ conv. 192 lReLU |
| $L - 3$ | $3 \times 3$ conv. 64 lReLU | $3 \times 3$ conv. 192 lReLU |
| $L - 2$ | $3 \times 3$ conv. 64 lReLU | $3 \times 3$ conv. 192 lReLU |
| $L - 1$ | global average pool | |
| $L - 0$ | 10 dense, softmax | |

Table 6: Small and Large CNN architectures. Leaky ReLU parameter $a = 0.1$. All convolutional and dense layers in the classifier are pre-activation batch-normalized. All images are resized to $32 \times 32 \times 3$. Note the use of additive Gaussian noise: this addition was motivated by initial experiments in which we observed that domain adversarial training appears to contract the feature space.

| Domain Discriminator |
|---|
| Layer $L - 5$ Output |
| 100 dense, ReLU |
| 1 dense, sigmoid |

Table 7: Domain discriminator architecture.

## B  HYPERPARAMETERS

We observed that extensive hyperparameter-tuning is not necessary to achieve state-of-the-art performance. To demonstrate this, we restrict our hyperparameter search for each task to $\lambda_d = \{0, 10^{-2}\}, \lambda_s = \{0, 1\}, \lambda_t = \{10^{-2}, 10^{-1}\}$, in all experiments with instance-normalized inputs. We fixed $\beta = 10^{-2}$. Note that the decision to turn $(\lambda_d, \lambda_s)$ on or off that can often be determined *a priori* based on prior belief regarding the extent to covariate shift. In the absence of such prior belief, a reliable choice is $(\lambda_d = 10^{-2}, \lambda_s = 1, \lambda_t = 10^{-2}, \beta = 10^{-2})$.

| Task | Instance-Normalized | $\lambda_d$ | $\lambda_s$ | $\lambda_t$ | $\beta$ |
|---|---|---|---|---|---|
| MNIST $\to$ MNIST-M | Yes, No | $10^{-2}$ | 0 | $10^{-2}$ | $10^{-2}$ |
| SVHN $\to$ MNIST | Yes, No | $10^{-2}$ | 0 | $10^{-2}$ | $10^{-2}$ |
| MNIST $\to$ SVHN | Yes | $10^{-2}$ | 1 | $10^{-2}$ | $10^{-2}$ |
| MNIST $\to$ SVHN | No | $10^{-2}$ | 1 | $10^{-2}$ | $10^{-3}$ |
| DIGITS $\to$ SVHN | Yes, No | $10^{-2}$ | 1 | $10^{-1}$ | $10^{-2}$ |
| SIGNS $\to$ GTSRB | Yes, No | $10^{-2}$ | 1 | $10^{-2}$ | $10^{-2}$ |
| CIFAR $\to$ STL | Yes, No | 0 | 1 | $10^{-1}$ | $10^{-2}$ |
| STL $\to$ CIFAR | Yes, No | 0 | 0 | $10^{-1}$ | $10^{-2}$ |
| Room A $\to$ B | Yes | 0 | 0 | $10^{-2}$ | $10^{-2}$ |

Table 8: Hyperparameters for each task, both with and without instance-normalized input. The only exception is MNIST $\to$ SVHN without instance-normalized input. In this specific case, $d_{\mathcal{H}\Delta\mathcal{H}}$ is sufficiently large that conditional entropy minimization quickly finds a degenerate solution in the target domain. To counter this, we remove conditional entropy minimization (but keep the target-side virtual adversarial training) *only* during VADA. We apply target-side conditional entropy minimization and virtual adversarial training during DIRT-T. To compensate, we use a lower $\beta$ during the DIRT-T phase to allow for larger natural gradient steps.

When the target domain is MNIST/MNIST-M, the task is sufficiently simple that we only allocate $B = 500$ iterations to each optimization problem in Eq. (14). In all other cases, we set the refinement interval $B = 5000$. We apply Adam Optimizer (learning rate $= 0.001, \beta_1 = 0.5, \beta_2 = 0.999$) with Polyak averaging (more accurately, we apply an exponential moving average with momentum $= 0.998$ to the parameter trajectory). VADA was trained for 80000 iterations and DIRT-T takes VADA as initialization and was trained for $\{20000, 40000, 60000, 80000\}$ iterations, with number of iterations chosen as hyperparameter.

## C  REPLACING GRADIENT REVERSAL

We note from Goodfellow et al. (2014) that the gradient of $\nabla_\theta \ln(1 - D(f_\theta(x)))$ is tends to have smaller norm than $-\nabla_\theta \ln D(f_\theta(x))$ during initial training since the latter rescales the gradient by $1/D(f_\theta(x))$. Following this observation, we replace the gradient reversal procedure with alternating minimization of

$$\min_D -\mathbb{E}_{x \sim \mathcal{D}_s}[\ln D(f_\theta(x))] - \mathbb{E}_{x \sim \mathcal{D}_t}[\ln 1 - D(f_\theta(x))]$$

$$\min_\theta -\mathbb{E}_{x \sim \mathcal{D}_t}[\ln D(f_\theta(x))] - \mathbb{E}_{x \sim \mathcal{D}_s}[\ln 1 - D(f_\theta(x))].$$

The choice of using gradient reversal versus alternating minimization reflects a difference in choice of approximating the mini-max using saturating versus non-saturating optimization (Fedus et al., 2017). In some of our initial experiments, we observed the replacement of gradient reversal with alternating minimization stabilizes domain adversarial training. However, we encourage practitioners to try either optimization strategy when applying VADA.

## D  INSTANCE NORMALIZATION FOR DOMAIN ADAPTATION

Theorem 1 suggests that we should identify ways of constraining the hypothesis space without hurting the global optimal classifier for the joint task. We propose to further constrain our model by

introducing instance normalization as an image pre-processing step for the input data. Instance normalization was proposed for style transfer Ulyanov et al. (2016) and applies the operation

$$\ell(x^{(i)}) = \frac{x^{(i)} - \mu(x^{(i)})}{\sigma(x^{(i)})}, \tag{15}$$

where $x^{(i)} \in \mathbb{R}^{H \times W \times C}$ denotes the $i^{\text{th}}$ sample with $(H, W, C)$ corresponding to the height, width, and channel dimensions, and where $\mu, \sigma : \mathbb{R}^{H \times W \times C} \to \mathbb{R}^C$ are functions that compute the mean and standard deviation across the spatial dimensions. A notable property of instance normalization is that it is invariant to channel-wide scaling and shifting of the input elements. Formally, consider scaling and shift variables $\gamma, \beta \in \mathbb{R}^C$. If $\gamma \succ 0$ and $\sigma(x^{(i)}) \succ 0$, then

$$\ell(x^{(i)}) = \ell(\gamma x^{(i)} + \beta). \tag{16}$$

For visual data the application of instance normalization to the input layer makes the classifier invariant to channel-wide shifts and scaling of the pixel intensities. For most visual tasks, sensitivity to channel-wide pixel intensity changes is not critical to the success of the classifier. As such, instance normalization of the input may help reduce $d_{\mathcal{H}\Delta\mathcal{H}}$ without hurting the globally optimal classifier. Interestingly, Figure 3 shows that input instance normalization is not equivalent to gray-scaling, since color is partially preserved. To test the effect of instance normalization, we report results both with and without the use of instance-normalized inputs.

## E  LIMITATION OF DOMAIN ADVERSARIAL TRAINING

We denote the source and target distributions respectively as $p_s(x, y)$ and $p_t(x, y)$. Let the source covariate distribution $p_s(x)$ define the random variable $X_s$ that have support $\text{supp}(X_s) = \mathcal{X}_s$ and let $(X_t, \mathcal{X}_t)$ be analogously defined for the target domain. Both $\mathcal{X}_s$ and $\mathcal{X}_t$ are subsets of $\mathbb{R}^n$. Let $p_s(y)$ and $p_t(y)$ define probabilities over the support $\mathcal{Y} = \{1, \dots, K\}$. We consider any embedding function $f : \mathbb{R}^n \to \mathbb{R}^m$, where $\mathbb{R}^m$ is the embedding space, and any embedding classifier $g : \mathbb{R}^m \to \mathcal{C}$, where $\mathcal{C}$ is the $(K-1)$-simplex. We denote a classifier $h = g \circ f$ has the composite of an embedding function with an embedding classifier.

For simplicity, we restrict our analysis to the simple case where $K = 2$, i.e. where $\mathcal{Y} = \{0, 1\}$. Furthermore, we assume that for any $\delta \in [0, 1]$, there exists a subset $\Omega \subseteq \mathbb{R}^n$ where $p_s(x \in \Omega) = \delta$. We impose a similar condition on $p_t(x)$.

For a joint distribution $p(x, y)$, we denote the generalization error of a classifier as

$$\epsilon_p(h) = \mathbb{E}_{p(x,y)} |y - h(x)|. \tag{17}$$

Note that for a given classifier $h : \mathbb{R}^n \to [0, 1]$, the corresponding hard classifier is $k(x) = \mathbb{1}\{h(x) > 0.5\}$. We further define the set $\Omega \subseteq \mathbb{R}^n$ such that

$$\Omega = \{x \in \mathbb{R}^n \mid k(x) = 1\} \iff k(x) = \mathbb{1}\{x \in \Omega\}. \tag{18}$$

In a slight abuse of notation, we define the generalization error $\epsilon(\Omega)$ with respect to $\Omega$ as

$$\epsilon_p(\Omega) = \mathbb{E}_{p(x,y)} \mathbb{1}\{x \in \Omega\} = \epsilon(k). \tag{19}$$

An optimal $\Omega_p^*$ is a partitioning of $\mathbb{R}^n$

$$\epsilon_p(\Omega_p^*) = \min_{\Omega \subseteq \mathbb{R}^n} \epsilon_p(\Omega) \tag{20}$$

such that generalization error under the distribution $p(x, y)$ is minimized.

### E.1  GOOD TARGET-DOMAIN ACCURACY IS NOT GUARANTEED

Domain adversarial training seeks to find a single classifier $h$ used for both the source $p_s$ and target $p_t$ distributions. To do so, domain adversarial training sets up the objective

$$\min_{f \in \mathcal{F}, g \in \mathcal{G}} \epsilon_{p_s}(g \circ f) \tag{21}$$

$$\text{s.t. } g(X_s) = g(X_t), \tag{22}$$

where $\mathcal{F}$ and $\mathcal{G}$ are the hypothesis spaces for the embedding function and embedding classifier. Intuitively, domain adversarial training operates under the hypothesis that good source generalization error in conjunction with source-target feature matching implies good target generalization error. We shall see, however, that if $\mathcal{X}_s \cap \mathcal{X}_t = \varnothing$ and $\mathcal{F}$ is sufficiently complex, this implication does not necessarily hold.

Let $\mathcal{F}$ contain all functions mapping $\mathbb{R}^n \to \mathbb{R}^m$, i.e. $\mathcal{F}$ has infinite capacity. Suppose $\mathcal{G}$ contains the function $g(z) = \mathbb{1}\{z = \mathbf{1}_m\}$ and $\mathcal{X}_s \cap \mathcal{X}_t = \varnothing$. We consider the set

$$\mathcal{H}^* = \left\{ g \circ f \mid \exists g \in \mathcal{G}, f \in \mathcal{F} \text{ s.t. } \epsilon_{p_s}(g \circ f) \leq \epsilon_{p_s}(\Omega_{p_s}^*), f(X_s) = f(X_t) \right\}. \tag{23}$$

Such a set of classifiers satisfies the feature-matching constraint while achieving source generalization error no worse than the optimal source-domain hard classifier. It suffices to show that $\mathcal{H}^*$ includes hypotheses that perform poorly in the target domain.

We first show $\mathcal{H}^*$ is not an empty set by constructing an element of this set. Choose a partitioning $\Omega$ where

$$p_t(x \in \Omega) = p_s(x \in \Omega_{p_s}^*). \tag{24}$$

Consider the embedding function

$$f_\Omega(x) = \begin{cases} \mathbf{1}_m & \text{if } (x \in \mathcal{X}_s \cap \Omega_{p_s}^*) \vee (x \in \mathcal{X}_t \cap \Omega) \\ \mathbf{0}_m & \text{otherwise.} \end{cases} \tag{25}$$

Let $g(z) = \mathbb{1}\{z = \mathbf{1}_m\}$. It follows that the composite classifier $h_\Omega = g \circ f_\Omega$ is an element of $\mathcal{H}^*$.

Next, we show that a classifier $h \in \mathcal{H}^*$ does not necessarily achieve good target generalization error. Consider the partitioning $\hat{\Omega}$ which solves the following optimization problem

$$\max_{\Omega \subseteq \mathbb{R}^n} \epsilon_{p_t}(\Omega) \tag{26}$$

$$\text{s.t. } p_t(x \in \Omega) = p_s(x \in \Omega_{p_s}^*). \tag{27}$$

Such a partitioning $\hat{\Omega}$ is the worst-case partitioning subject to the probability mass constraint. It follows that worse case $h' \in \mathcal{H}^*$ has generalization error

$$\epsilon_{p_t}(h') = \max_{h \in \mathcal{H}^*} \epsilon_{p_t}(h) \geq \epsilon_{p_t}(h_{\hat{\Omega}}). \tag{28}$$

To provide intuition that $\epsilon_{p_t}(h')$ is potentially very large, consider hypothetical source and target domains where $\mathcal{X}_s \cap \mathcal{X}_t = \varnothing$ and $p_t(x \in \Omega_{p_t}^*) = p_s(x \in \Omega_{p_s}^*) = 0.5$. The worst-case partitioning subject to the probability mass constraint is simply $\hat{\Omega} = \mathbb{R}^n \setminus \Omega_{p_t}^*$ (which flips the labels) and consequently, $\mathcal{H}^*$ contains solutions

$$\max_{h \in \mathcal{H}^*} \epsilon_{p_t}(h) \geq 1 - \epsilon_{p_t}(\Omega_{p_t}^*) \tag{29}$$

no better than the worst-case partitioning of the target domain.

### E.2 CONNECTION TO THEOREM 1

Let $\mathcal{F}$ contain all functions mapping $\mathbb{R}^n \to \mathbb{R}^m$, i.e. $\mathcal{F}$ has infinite capacity. Suppose $\mathcal{G}$ contains the function $g(z) = \mathbb{1}\{z = \mathbf{1}_m\}$ and $\mathcal{X}_s \cap \mathcal{X}_t = \varnothing$. We consider the sets

$$\mathcal{H} = \{ g \circ f \mid g \in \mathcal{G}, f \in \mathcal{F} \} \tag{30}$$

$$\bar{\mathcal{H}} = \{ g \circ f \mid \exists g \in \mathcal{G}, f \in \mathcal{F} \text{ s.t. } f(X_s) = f(X_t) \}. \tag{31}$$

A justification for domain adversarial training is that the $\bar{\mathcal{H}} \Delta \bar{\mathcal{H}}$-divergence term is smaller than the $\mathcal{H} \Delta \mathcal{H}$-divergence, thus yielding a tighter upper bound for Theorem 1. However, we shall see that the $\mathcal{H} \Delta \bar{\mathcal{H}}$-divergence term is in fact maximal.

Choose partitionings $\Omega_s, \Omega_t \subseteq \mathbb{R}^n$ such that

$$p_s(x \in \Omega_s) = p_t(x \in \Omega_t) = 0.5. \tag{32}$$

Define the embedding functions

$$f(x) = \begin{cases} \mathbf{1}_m & \text{if } (x \in \mathcal{X}_s \cap \Omega_s) \vee (x \in \mathcal{X}_t \cap \Omega_t) \\ \mathbf{0}_m & \text{otherwise.} \end{cases} \tag{33}$$

$$f'(x) = \begin{cases} \mathbf{1}_m & \text{if } (x \in \mathcal{X}_s \cap \Omega_s) \vee (x \in \mathcal{X}_t \cap (\mathbb{R}^n \setminus \Omega_t)) \\ \mathbf{0}_m & \text{otherwise.} \end{cases} \tag{34}$$

Let $g'(z) = g(z) = \mathbb{1}\{z = \mathbf{1}_m\}$. It follows that the composite classifiers $h = g \circ f$ and $h' = g' \circ f'$ are elements of $\bar{\mathcal{H}}$.

From the definition of $d_{\mathcal{H}\Delta\mathcal{H}}$, we see that

$$d_{\bar{\mathcal{H}}\Delta\bar{\mathcal{H}}} \geq 2\left| \mathbb{E}_{x \sim X_s} [h(x) \neq h'(x)] - \mathbb{E}_{x \sim X_t} [h(x) \neq h'(x)] \right| \tag{35}$$
$$= 2 \cdot |0 - 1| = 2. \tag{36}$$

The $\bar{\mathcal{H}}\Delta\bar{\mathcal{H}}$-divergence thus achieves the maximum value of 2.

### E.3    IMPLICATIONS

Our analysis assumes infinite capacity embedding functions and the ability to solve optimization problems exactly. The empirical success of domain adversarial training suggests that the use of finite-capacity convolutional neural networks combined with stochastic gradient-based optimization provides the necessary regularization for domain adversarial training to work. The theoretical characterization of domain adversarial training in the case finite-capacity convolutional neural networks and gradient-based learning remains a challenging but important open research problem.

## F    NON-VISUAL DOMAIN ADAPTATION TASK

To evaluate the performance of our models on a non-visual domain adaptation task, we applied VADA and DIRT-T to the Wi-Fi Activity Recognition Dataset (Yousefi et al., 2017). The Wi-Fi Activity Recognition Dataset is a classification task that takes the Wi-Fi Channel State Information (CSI) data stream as input $x$ to predict motion activity within an indoor area as output $y$. The dataset collected the CSI data stream samples associated with seven activities, denoted as "bed", "fall", "walk", "pick up", "run", "sit down", and "stand up".

However, the joint distribution over the CSI data stream and motion activity changes depending on the room in which the data was collected. Since the data was collected for multiple rooms, we selected two rooms (denoted here as Room A and Room B) and constructed the unsupervised domain adaptation task by using Room A as the source domain and Room B as the target domain. We compare the performance of DANN, VADA, and DIRT-T on the Wi-Fi domain adaptation task in Table 2, using the hyperparameters $(\lambda_d = 0, \lambda_s = 0, \lambda_t = 10^{-2}, \beta = 10^{-2})$.

Table 2 shows that VADA significantly improves classification accuracy compared to Source-Only and DANN. However, DIRT-T does not lead to further improvements on this dataset. We believe this is attributable to VADA successfully pushing the decision boundary away from data-dense regions in the target domain. As a result, further application of DIRT-T would not lead to better decision boundaries. To validate this hypothesis, we visualize the t-SNE embeddings for VADA and DIRT-T in Figure 6 and show that VADA is already capable of yielding strong clustering in the target domain. To verify that the decision boundary indeed did not change significantly, we additionally provide the confusion matrix between the VADA and DIRT-T predictions in the target domain (Fig. 7).

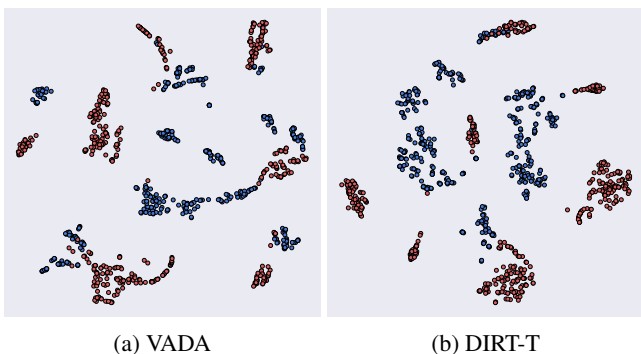

(a) VADA       (b) DIRT-T

Figure 6: T-SNE plot of the last hidden layer for Room A (blue) $\rightarrow$ Room B (red)

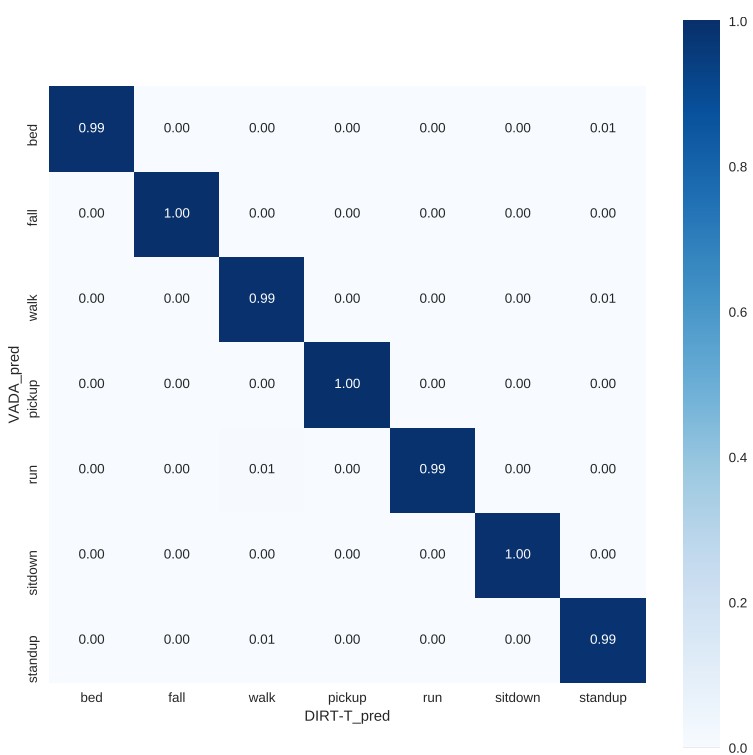

Figure 7: Confusion matrix between VADA and and DIRT-T prediction labels.

