# OpenReview forum: "A DIRT-T Approach to Unsupervised Domain Adaptation"
_ICLR.cc/2018/Conference — Accept (Poster)_

### Official Review · AnonReviewer3 · 2017-11-27
**A sound approach to mix two complementary strategies for domain adaptation**

**Rating:** 7
**Confidence:** 4

**Review:**

As there are many kinds of domain adaptation problems, the need to mix several learning strategies to improve the existing approaches is obvious. However, this task is not necessarily easy to succeed. The authors proposed a sound approach to learn a proper representation (in an adversarial way) and comply the cluster assumption.

The experiments show that this Virtual Adversarial Domain Adaptation network (VADA) achieves great results when compared to existing learning algorithms. Moreover, we also see the learned model is consistently improved using the proposed "Decision-boundary Iterative Refinement Training with a Teacher" (DIRT-T) approach.

The proposed methodology relies on multiple choices that could sometimes be better studied and/or explained. Namely, I would like to empirically see which role of the locally-Lipschitz regularization term (Equation 7). Also, I wonder why this term is tuned by an hyperparameter (lamda_s) for the source, while a single hyperparamer (lambda_t) is used for the sum of the two target quantity.

On the theoretical side, the discussion could be improved. Namely, Section 3 about "limitation of domain adversarial training" correctly explained that "domain adversarial training may not be sufficient for domain adaptation if the feature extraction function has high-capacity". It would be interesting to explain whether this observation is consistent with Theorem 1 of the paper (due to Ben-David et al., 2010), on which several domain adversarial approaches are based. The need to consider supplementary assumptions (such as ) to achieve good adaptation can also be studied through the lens of more recent Ben-David's work, e.g. Ben-David and Urner (2014). In the latter, the notion of "Probabilistic Lipschitzness", which is a relaxation of the "cluster assumption" seems very related to the actual work.

Reference:
Ben-David and Urner. Domain adaptation-can quantity compensate for quality?, Ann. Math. Artif. Intell., 2014

Pros:
- Propose a sound approach to mix two complementary strategies for domain adaptation.
- Great empirical results.

Cons:
- Some choices leading to the optimization problem are not sufficiently explained.
- The theoretical discussion could be improved.

Typos:
- Equation 14: In the first term (target loss), theta should have an index t (I think).
- Bottom of page 6: "... and that as our validation set" (missing word).

---

> ### Author Response · Authors · 2018-01-05
> **Response to Reviewer 3**
>
> Thank you for the review! To improve the quality of the paper, we have made several adjustments to our paper in accordance with your review.
>
> “Namely, I would like to empirically see which role of the locally-Lipschitz regularization term (Equation 7).”
>
> Thank you for the suggestion. We have included an extensive ablation study of the role of the locally-Lipschitz regularization term in Section 6.3.1. Our results show that while conditional entropy minimization alone is sufficient to instantiate the cluster assumption and improve over DANN, the additional incorporation of the locally-Lipschitz regularization term does indeed offer additional performance improvement.
>
> “Also, I wonder why this term is tuned by an hyperparameter (λ_s) for the source, while a single hyperparamer (λ_t) is used for the sum of the two target quantity”
>
> The choice to use λ_t for the sum of the two target quantities is purely for simplicity. Since the official implementation of VAT (https://github.com/takerum/vat_tf) used the same weighting for the conditional entropy and virtual adversarial training, we opted to do that as well in the target domain.
>
> “It would be interesting to explain whether this observation is consistent with Theorem 1 of the paper (due to Ben-David et al., 2010)”
>
> Thank you for the suggestion. We have added the connection in Appendix E. In particular, we can show that, if the embedding function has infinite-capacity, the H\DeltaH-divergence achieves the maximum value of 2 even when the feature distribution matching constraint is satisfied. This results in Theorem 1 becoming a vacuous upper bound.
>
> “The notion of ‘Probabilistic Lipschitzness’, which is a relaxation of the ‘cluster assumption’ seems very related to the actual work.”
>
> Thank you for this insight. We have incorporated a brief mention of probabilistic Lipschitzness in Section 2. It seems that a stronger connection can be made and we appreciate any additional suggestions you may have on how to better address probabilistic Lipschitzness in our paper.
>
> “Equation 14: In the first term (target loss), theta should have an index t (I think).” and “Bottom of page 6: ‘... and that as our validation set’ (missing word).”
>
> Fixed. Thanks!

---

> > ### Comment · AnonReviewer3 · 2018-01-12
> > **A satisfied mind**
> >
> > I'm truly satisfied by the new experiments, which demonstrate the benefit of "virtual adversarial training" in the whole process.
> >
> > Unfortunately, I did not take the time to explore the connection of "Probabilistic Lipschitzness" with the current work, but it's certainly an interesting thing to look at.

---

### Official Review · AnonReviewer1 · 2017-11-27

**Rating:** 8
**Confidence:** 4

**Review:**

This paper presents two complementary models for unsupervised domain adaptation (classification task): 1) the Virtual Adversarial Domain Adaptation (VADA) and 2) the Decision-boundary Iterative Refinement Training with a Teacher (DIRT-T). The authors make use of the so-called cluster assumption, i.e., decision boundaries should not cross high-density data regions. VADA extends the standard Domain-Adversarial training by introducing an additional objective L_t that measures the target-side cluster assumption violation, namely, the conditional entropy w.r.t. the target distribution. Since the empirical estimate of the conditional entropy breaks down for non-locally-Lipschitz classifiers, the authors also propose to incorporate virtual adversarial training in order to make the classifier well-behaved. The paper also argues that the performance on the target domain can be further improved by a post-hoc minimization of L_t using natural gradient descent (DIRT-T) which ensures that the decision boundary changes incrementally and slowly.

Pros:
+ The paper is written clearly and easy to read
+ The idea to keep the decision boundary in the low-density region of the target domain makes sense
+ The both proposed methods seem to be quite easy to implement and incorporate into existing DATNN-based frameworks
+ The combination of VADA and DIRT-T performs better than existing DA algorithms on a range of visual DA benchmarks

Cons:
- Table 1 can be a bit misleading as the performance improvements may be partially attributed to the fact that different methods employ different base NN architectures and different optimizers
- The paper deals exclusively with visual domains; applying the proposed methods to other modalities would make this submission stronger

Overall, I think it is a good paper and deserves to be accepted to the conference. I’m especially appealed by the fact that the ideas presented in this work, despite being simple, demonstrate excellent performance.

Post-rebuttal revision:
After reading the authors' response to my review, I decided to leave the score as is.

---

> ### Author Response · Authors · 2018-01-05
> **Response to Reviewer 1**
>
> Thank you for the review! To improve the quality of the paper, we have made several adjustments to our paper in accordance with your review.
>
> “Table 1 can be a bit misleading as the performance improvements may be partially attributed to the fact that different methods employ different base NN architectures and different optimizers”
>
> The purpose of Table 1 is to offer a holistic evaluation of the entire set-up. As such, it demonstrates that there exists a training objective + architecture + optimization configuration such significant improvements over previous methods/implementations are possible. We provided such a table in part because doing so seems to be standard practice in semi-supervised learning and domain adaptation papers (Miyato et al., 2017; Laine & Aila, 2016; Tarvainen & Valpola, 2017; Saito et al., 2017; French et al., 2017).
>
> To offer a fairer comparison, we made the following modifications to the paper:
>
> 1. We explicitly mention Table 2 in the main body of the paper, in the section Model Evaluation/Overall
>
> 2. We added an ablation study to Section 6.3.1 to demonstrate the relative contribution of virtual adversarial training in comparison to our base implementation of domain adversarial training.
>
> “The paper deals exclusively with visual domains; applying the proposed methods to other modalities would make this submission stronger”
>
> We agree that the submission would be stronger by performing experiments in other modalities. To that end, we added an example of applying VADA and DIRT-T to a non-visual data in Section 6.2. We chose to apply our model to a Wi-Fi activity recognition dataset. Our results show that VADA significantly improves upon DANN. Unfortunately, due to the small target domain training set size, DIRT-T does not improve upon VADA. We provide additional experiments in Appendix F which suggest that VADA already achieves strong clustering on the Wi-Fi dataset, and therefore DIRT-T is not expected to improve performance in this situation.
>
> We leave as future work the study of applying VADA/DIRT-T (and the general application of the cluster assumption) to text classification domain adaptation tasks. Given the success of VAT on text classification (Miyato et al., 2016), we are optimistic that this line of work is promising.
>
> References
> Takeru Miyato, Shin-ichi Maeda, Masanori Koyama, and Shin Ishii. Virtual adversarial train- ing: a regularization method for supervised and semi-supervised learning. arXiv preprint arXiv:1704.03976, 2017.
>
> Takeru Miyato, Andrew M Dai, and Ian Goodfellow. Virtual adversarial training for semi-supervised text classification. stat, 1050:25, 2016.
>
> Samuli Laine and Timo Aila. Temporal ensembling for semi-supervised learning. arXiv preprint arXiv:1610.02242, 2016.
>
> Antti Tarvainen and Harri Valpola. Mean teachers are better role models: Weight-averaged consis- tency targets improve semi-supervised deep learning results. 2017.
>
> Geoffrey French, Michal Mackiewicz, and Mark Fisher. Self-ensembling for domain adaptation. arXiv preprint arXiv:1706.05208, 2017.
>
> Kuniaki Saito, Yoshitaka Ushiku, and Tatsuya Harada. Asymmetric tri-training for unsupervised domain adaptation. arXiv preprint arXiv:1702.08400, 2017.

---

### Official Review · AnonReviewer2 · 2017-12-05
**Good contribution to unsupervised domain adaptation**

**Rating:** 7
**Confidence:** 2

**Review:**

The paper was a good contribution to domain adaptation. It provided a new way of looking at the problem by using the cluster assumption. The experimental evaluation was very thorough and shows that VADA and DIRT-T performs really well.

I found the math to be a bit problematic. For example, L_d in (4) involves a max operator. Although I understand what the authors mean, I don't think this is the correct way to write this. (5) should discuss the min-max objective. This will probably involve an explanation of the gradient reversal etc. Speaking of GRL, it's mentioned on p.6 that they replaced GRL with the traditional GAN objective. This is actually pretty important to discuss in detail: did that change the symmetric nature of domain-adversarial training to the asymmetric nature of traditional GAN training? Why was that important to the authors?

The literature review could also include Shrivastava et al. and Bousmalis et al. from CVPR 2017. The latter also had MNIST/MNIST-M experiments.

---

> ### Author Response · Authors · 2018-01-05
> **Response to Reviewer 2**
>
> Thank you for the review! To improve the quality of the paper, we have made several adjustments to our paper in accordance with your review.
>
> “The experimental evaluation was very thorough and shows that VADA and DIRT-T performs really well.”
>
> We have added additional experiments that may be of interest to you. In Section 6.3.1, we perform an extensive ablation study demonstrate the relative contribution of virtual adversarial training. In Section 6.2, we apply VADA/DIRT-T to a non-visual domain adaptation task.
>
> “For example, L_d in (4) involves a max operator. Although I understand what the authors mean, I don’t think this is the correct way to write this.”
>
> We agree the the use of the max operator is informal. To account for the possibility that the maximum is not achievable, using the supremum is more appropriate. We have updated the paper to reflect this. Our choice of presentation is now in keeping with that in GAIL (Eq. (14), Ho & Ermon (2016)) and WGAN (Eq. (2), Arjovsky et al. (2017)).
>
> “(5) should discuss the min-max objective. This will probably involve an explanation of the gradient reversal etc. Speaking of GRL, it’s mentioned on p.6 that they replaced GRL with the traditional GAN objective. This is actually pretty important to discuss in detail: did that change the symmetric nature of domain-adversarial training to the asymmetric nature of traditional GAN training? Why was that important to the authors?”
>
> Thank you for pointing this out. We have added a footnote next to (5) and modified Appendix C to reflect the following opinion:
>
> We believe that, at a high level, it is not of particular importance which optimization procedure is used to approximate the mini-max optimization problem. Our decision to switch from the symmetric to asymmetric training is motivated by
>
> 1. The extensive GAN literature which advocates the asymmetric optimization approach.
>
> 2. Our initial experiments on MNIST → MNIST-M which suggest that the asymmetric optimization approach is more stable.
>
> We are not committed to either optimization strategy and encourage practitioners to try both when applying VADA. In case the reviewer is interested in the performance of pure domain adversarial training using the asymmetric optimization approach, we have included it in Section 6.3.1.
>
> “The literature review could also include Shrivastava et al. and Bousmalis et al. from CVPR 2017. The latter also had MNIST/MNIST-M experiments.”
>
> Thank you for the suggestion. We have incorporated Bousmalis’s paper into our comparison.
>
> References
> Jonathan Ho and Stefano Ermon. Generative adversarial imitation learning. In Advances in Neural Information Processing Systems, pp. 4565–4573, 2016.
>
> Martin Arjovsky, Soumith Chintala, and Le ́on Bottou. Wasserstein gan. arXiv preprint arXiv:1701.07875, 2017.

---

### Author Response · Authors · 2018-01-05
**Revisions to paper**

To improve the quality of the paper, we have made several adjustments to our paper. In addition to minor edits (e.g. fixing typos, improving clarity), we made the following large changes:

1. We added a non-visual domain adaptation task (Wi-Fi activity recognition) to Section 6.2 and Appendix F.

2. We added an additional ablation experiment testing the contribution of virtual adversarial training to Section 6.3.

3. We improved the presentation of Proposition 1 in Appendix E and added a subsection connecting Proposition 1 to Ben-David’s domain adaptation upper bound (Theorem 1).

Thank you for taking the time to review this paper!

---

### Decision · Program_Chairs · 2018-01-29
**ICLR 2018 Conference Acceptance Decision**

**Decision:**

Accept (Poster)

**Comment:**

Well motivated and well written, with extensive results. The paper also received positive comments from all reviewers. The AC recommends that the paper be accepted.